# “Instead of Being on a Screen You Can Be More Out There and Enjoy Your Life”: Young People’s Understandings of Physical Activity for Health

**DOI:** 10.3390/ijerph20105880

**Published:** 2023-05-19

**Authors:** Natasha Wilson, Lorraine Cale, Ashley Casey

**Affiliations:** School of Sport, Exercise and Health Sciences, Loughborough University, Loughborough LE11 3TU, UK; l.a.cale@lboro.ac.uk (L.C.); a.j.b.casey@lboro.ac.uk (A.C.)

**Keywords:** physical activity, physical education, health, digital technologies, young people, active lifestyles, social media

## Abstract

Background: Despite documented evidence of the benefits of leading a physically active life, it is reported that less than half of young people in Europe meet the physical activity recommendations. Schools, and in particular physical education (PE), are viewed to be at the forefront of addressing inactive lifestyles and educating young people about physical activity. Nonetheless, given advancements in technology, young people are increasingly exposed to physical activity information “beyond the school gates”. Consequently, if PE teachers are to support young people to understand the information they receive surrounding physical activity online, then they need to be able to address any misconceptions about health they may have. Methods: In this study, fourteen young people (7 boys and 7 girls) in year 9 (13–14 years old) from two secondary schools in England participated in a digitally-based activity and semi-structured interviews which aimed to explore their conceptions of physical activity for health. Results: It was found that the young people had limited and narrow conceptions of what it means to be physically active. Conclusions: It was suggested the findings could be partly attributed to limitations in students’ learning and experiences with respect to physical activity and health in the PE curriculum.

## 1. Introduction

Despite the documented benefits of leading a physically active life [1,2], evidence suggests that less than half of young people in all countries across Europe meet the physical activity recommendations of 60 min of moderate to vigorous daily exercise [3]. Globally it is estimated that only 31% of young people under the age of 15 engage in sufficient physical activity [4]. The decline in jobs requiring manual labour, coupled with the ready accessibility of cars in many households, means modern youth are the first generation who are required to make a conscious decision to factor physical activity into their lives [2].

It has been acknowledged previously that schools and physical education (PE) have a role to play in the promotion of a healthy active lifestyle [5,6]. Yet, evidence suggests that education and PE have had little impact on daily activity levels [7]. Consequently, it is important that schools and PE foster young people’s understanding of physical activity and health and PE teachers address any misconceptions their students may have regarding what leading a physically active lifestyle entails. This is particularly important given: (a)Lifelong physical activity behaviours are formed in childhood [8,9,10];(b)An inactive childhood is linked to a sedentary lifestyle in adulthood [11,12];(c)The associated health complications of inactive lifestyles cost Britain an estimated £7.4 billion annually [2];(d)Globally sedentary lifestyles result in 3.2 million deaths annually [13].

### 1.1. Young People’s Understandings of Physical Activity for Health

Previous research has highlighted concerns regarding young people’s knowledge and understanding of health and physical activity concepts [14,15,16,17]. Indeed, studies conducted in a number of countries (e.g., in the United Kingdom (UK), New Zealand and United States (US)) have suggested that young people hold a simplistic and narrow understanding of health and physical activity [14,15,17,18,19,20,21,22,23,24]. The early work by Harris [15] found that young people defined health exclusively in a physical sense and held narrow and limited conceptions of what it meant to be healthy, equating health with exercise but failing to recognise the social and psychological benefits of being active. Similarly, Burrows [18] and Burrows and Wright [24] reported that when students talked about health they equated it with weight, body size and shape. These authors concluded that New Zealand children’s ideas of health were perceived predominantly in a corporeal sense. 

Placek et al.’s [17] research with middle school students in the US found that youngsters did not understand the purpose of fitness, equating it with one’s appearance, and were unable to identify appropriate exercises to improve different types of fitness. Placek et al. [17] concluded that students had a surface familiarity with different exercises such as jogging, but did not understand the purpose of and the distinction between exercise and fitness. Importantly, given their findings, Harris et al. [20] reported that young people’s confusion and/or misunderstanding about health concepts could potentially hinder physical activity promotion. This research raises questions regarding how and where young people are developing such misunderstandings and the role and effectiveness of schools and PE in addressing them and in promoting their knowledge and understanding of health and physical activity.

### 1.2. Digital Technologies, Social Media and Physical Activity for Health Knowledge

Studies conducted in the UK, Europe and the US show that two-thirds of young people (average age 14 years) use social media for a minimum of three hours daily and one-fifth use social media for more than five hours daily [25,26,27]. Given the increasing prevalence and accessibility of digital technologies and social media [25], there has been a shift in contemporary society whereby young people can easily access health information outside of the classroom [28]. This increase in the availability of health and physical activity digital information, coupled with a lack of quality checking, means that young people could be subjected to messages and information that are conflicting, contrasting or even incorrect [20,21]. Consequently, schools and PE face the challenge not only of promoting and developing young people’s understanding of physical activity and health and supporting them to lead a physically active lifestyle, but also of confronting any misinformation young people access or receive through digital technologies. Ultimately, this will require PE teachers to support and equip young people with the skills to critique and become critical consumers of health information [29] and of the images they source through social media and other digital mediums. 

To be able to do this, PE teachers firstly need to be aware of the extent to which young people rely on social media to source health and physical activity information and to recognise the influence of this on their physical activity knowledge and behaviours. Only then will schools be able to help young people to critically assess the messages they receive about health and physical activity. The purpose of this paper, therefore, is to explore young people’s understandings of what it means to be physically active.

## 2. Methods

### 2.1. Study Participants, Setting, and Design

This study was part of a larger research project focused on digital public pedagogies and physical activity for health information in young people. The participants were 14 young people (7 boys and 7 girls) in year 9 (aged 13–14 years of age) from two different secondary schools in England. This age group was selected in the inclusion criteria because 13 is the minimum legal age in the UK to be able to register for the majority of social media accounts. The exclusion criteria included anyone who was unwilling to give informed consent or assent. Participants were recruited from partnership schools which were involved in Initial Teacher Training (ITT) with a university in central England. An invitation email was sent to each school’s ITT coordinator, i.e., the senior teacher with responsibility for all student teachers placed in the school. This email contained information about the study and inclusion criteria and requested that participants include a mixture of males and females from a range of social backgrounds. Prior to data collection, ethical approval for the study was obtained from the university’s ethics approvals (human participants) sub-committee. Formal consent to conduct the study was sought and granted from the gatekeepers in each of the schools (headteachers) and the legal guardians of the young people. Assent was sought and gained from the young people themselves. The study involved two phases: a digitally-based activity and semi-structured interviews. 

### 2.2. The COVID-19 Pandemic

It is important to note that data gathering was directly impacted by the COVID-19 pandemic. This meant that the study was forced online after the completion of data gathering with school 1 (participants 1–6). At this time, we were aware of the new COVID-19 virus that was affecting China, but did not predict the global impact the virus would have. Consequently, while data gathering with school 1 occurred prior to the first COVID-19 lockdown in the UK (12 February to mid-March 2020), participants from the second school (participants 7–14) undertook the two phases of data gathering online due to the restrictions mandated during this period. While this led to a hybrid data gathering process, we felt it honored the voices and commitment of both groups of participants. It is also important to note that the move to online schooling had an impact on anticipated participant numbers. 

### 2.3. Digitally-Based Activity

In phase 1, an interactive activity was conducted with the young people using the social media platform Pinterest (Pinterest LTD California, 2022) and facilitated by the first author. Each student was provided with a tablet (Huawei Media Pad T3 _7_, Model: BG2-W09) and an individual research account was created for them on Pinterest using a unique email address (for example (UNINAME1001@gmail.com). The students were asked to work on the tablets individually to find images that they believed represented ‘the benefits of being physically active for an active lifestyle’ and to pin them to their Pinterest board. The digitally-based activity was completed during normal school hours at a mutually convenient time for the schools, and on average took 30–45 min to complete. Upon completion of the task the researcher saved the boards in order that they could be drawn on during phase 2 of the study and the young people were briefed regarding the nature of the follow up interview. 

### 2.4. Semi-Structured Interviews 

Semi-structured interviews were conducted with each participant one week after they completed the digitally-based activity. The interview schedule was designed to promote a conversation around the Pinterest boards the young people had created and to explore their conceptions of physical activity for health. For example, questions included: What made you choose these images? What images on Pinterest did you see that were to do with the benefits of being physically active and leading an active lifestyle? Did you avoid choosing any images on Pinterest? And if so, why? All questions were open ended to encourage participants to share their thoughts, ideas and feelings and in an effort to avoid bias by suggesting responses, which is a criticism of closed questions [30]. To gain further clarification, and to encourage more in depth responses from the interviewees, probing questions were also used, as appropriate, such as, Can you provide an example? Can you explain further? The interviews were held during normal school hours at a mutually convenient time for class teachers and recorded using a Sony digital dictation machine (ICD-Px470), with each taking between 12 and 30 min. 

### 2.5. Data Analysis

All interviews were transcribed verbatim as soon as possible after the interview. Following this, each transcript was loaded on the NVivo (version 12) software package. The data were then thematically analysed using Braun and Clarke’s [31] six steps to thematic analysis: familiarisation with the data; generating codes; searching for themes; reviewing themes; defining and categorising themes; and writing up the data. Due to limited access to the university’s facilities during the COVID-19 pandemic, and therefore to the first author’s university computer, some coding was completed manually prior to this being uploaded to NVivo.

The initial step of data analysis involved the first author familiarising herself with the data by reading and re-reading the transcripts and making notes of preliminary ideas. Initial codes were then generated through reading the interview data and identifying sections of text that were important to the research questions. Thirdly, coding through NVivo decreased the volume of the raw data which left manageable sections of text to allocate into themes [32]. Step four involved reviewing the themes and step five checking each individual theme in detail and deciding whether complex themes required structuring with sub-themes. The final step involved writing up the data extracts based on the analysis and relating this back to the literature. Although the six steps of this thematic analysis are presented as a linear process, in practice the analysis was iterative with movement back and forth between the different steps [33]. Once the initial codes and themes had been developed, all three authors re-examined the coded data and themes and verified them by consensus. Three key themes identified and constructed by the researchers from the qualitative data were: (1) sport as physical activity for health; (2) body shape and a balanced diet; and (3) mental health, getting out and friendship. 

## 3. Results and Discussion

### 3.1. Sport as Physical Activity for Health 

When asked to select and explain images that represented what they believed physical activity for health was, the majority of young people chose only images of sports and spoke only about sports during their interview. The fact that young people predominantly equated physical activity with sport is not incorrect. It does, however, represent a narrow and limited interpretation of physical activity. Physical activity extends beyond sport to include, for example, active recreation, walking, cycling and dance [34], and yet most of the participants did not identify or refer to any of these or other similar activities in the study. One young person, for instance, noted, “being physically active is like playing sports in my opinion, because I like playing sports so that’s what keeps me fit and all that stuff”. Similarly, another young person considered health to be “like sports and things because that’s what I think associates with healthy lifestyle”. 

Given the images the participants chose in the digital task (see Figure 1 for some examples) and their responses, it may be that the young people’s prior experiences were narrow in focus. Whilst the hope in setting this task was that the young people would be able to draw on their prior experiences both within school (including in PE and extra-curricular activities) and beyond the school gates (e.g., in the local community, media, etc.), it seems that the majority had a restrictive view of physical activity, associating it predominantly with sports.

In spite of repeated calls for schools and PE to offer a range of physical activities and a broad and balanced curriculum [35,36,37], literature suggests the subject has been, and still is, dominated by sport [38,39]. According to Quennerstedt [37] (p. 5), PE has predominantly focused on sporting techniques and ball games”, whereas others have claimed that it has been dominated by ‘fitness’ for sports performance or learning specific sporting drills [5,40,41]. Either way, the young people’s perceptions that physical activity for health is best represented by sport suggest that their learning about wider physical activity and positive health behaviours has perhaps been overshadowed by sporting drills and techniques. It would seem PE teachers still have some work to do in addressing the above conceptions through the curriculum they offer and the messages they promote.

For many participants, not only were physical activity for health and sport synonymous, but sport was also the main vehicle for staying or becoming healthy. One participant commented, “sport is about keeping healthy and not just showing off if you know what I mean”. A second explained that the sports images they selected “kind of shows like the work they [the athletes in the pictures] have gone through to get to where they are”, and recognised how “the fact is they have sports and how they end up doing sports shows the benefits you can have after hard work in sports”. 

The benefits of sports participation are well documented by scholars and policy makers alike, ranging from the psychological to the physiological [42]. Indeed, the contribution of sport towards positive health outcomes is arguably the primary justification for governments subsidising sport, and for continuing to promote its benefits to society [43]. The recognised health benefits of sport are also cited in the WHO’s 2013 [44] and 2018 [45] Global Action Plans as a key intervention in preventing non-communicable diseases [43]. Despite this, there appears to be an association in some of the data in this study between fandom, admiration, and physical activity for health. In their work exploring the difficulties for professional sports organisations in balancing the promotion of health with the delivery of social and business outcomes, Hills, Walker and Barry [42] concluded that there could be no outright winner. As such, the association made by some of the young people between aspiration and physical activity for health is, in all likelihood, more the product of marketing than social change. 

### 3.2. Body Shape and a Balanced Diet 

#### 3.2.1. Body Shape

Regardless of the broad brief the research participants were given to select images that represented being physically active for health, a number of the images selected also seemed to reveal a pre-occupation with body shape, weight and dieting. A range of images were shared which depicted young, slim individuals (see Figure 2), and included illustrations such as how to lose weight on targeted areas of the body and an advert for an application that promotes weight loss in 14 days. 

The selection of such images suggests that many of the participants conflated physical activity and health with body shape and weight. The concern here is that young people could begin to compare themselves against unrealistic appearance ideals which, in turn, could lead to body dissatisfaction [46,47] as well as to unhealthy practices/habits. In their work on body image, Grogan [48] found body dissatisfaction in male and female adolescents to correlate with unhealthy weight management strategies, such as reducing calorie intake to low levels while increasing the volume of exercise in efforts to achieve a desirable body shape. Similarly, Tiggeman and Slater [49] explored the relationship between body image concerns and exposure to the Internet, and found that the Internet can lead to body self-surveillance and the internalisation of the thin ideal. Meanwhile, Dion et al. [50] held that the high prevalence of body dissatisfaction in young people can potentially be exacerbated by the messages and images of beauty ideals they are increasingly seeing earlier in life, in general, on social media. Data from the current study suggests similar signifiers of health are prevalent and persist. 

Due to the subject’s focus on the physical body, body shape has been a dominant topic of debate in PE research for some time (see [14,17,20,21,51,52,53]). Indeed, as early as the beginning of this century, Evans [54] highlighted how success in PE was being defined by size, shape and weight. Other research has reported similar conflations between body weight and shape and key health-related concepts. For example, in Burrows and colleagues’ [14] study of New Zealand children’s constructions of health and fitness, the youngsters viewed body shape to be a gauge of fitness, and notably also believed weighing scales to be crucial in any fitness plan.

More recently, Barker et al. [55] have demonstrated the significant potential PE has in constructing how young people view their own body and the bodies of others. These authors identified schools and PE as contexts in which to improve young people’s perceptions of their own body. Despite recognising the potential, Barker et al. [55] equally acknowledged how the subject can have an adverse impact on some students who learn to see their bodies in a negative way. Similar concerns regarding how a focus on body image in PE can have negative consequences and lead to body dissatisfaction have been raised by others [56,57,58]. Indeed, Cale and Harris [56] warned of the effects of government policies and the dominance of health and obesity discourses on health education and PE in schools which focus attention on ‘corporeal perfection’ or the “slender ideal” (p. 440) and where measurement, assessment and comparison in young people are celebrated. 

Given the above, and the images the participants selected, it would seem PE, and schools in general, need to carefully consider how they address issues such as weight management and how they frame messages relating to weight, body image and shape. It is argued that less emphasis needs to be placed on measuring, body mass index and fitness testing young people in PE [59], and more on involving them in meaningful physical activity. Ultimately there remains a need to end the current and seemingly well held notion that being thin = healthy and being fat = unhealthy [56,60].

#### 3.2.2. A Balanced Diet

Despite the focus of the task being physical activity and an active lifestyle, it was also noteworthy that many of the young people selected images relating to nutrition and/or healthy eating. When asked why, one young participant explained “it is important to eat healthy and make sure your body is getting all the stuff it needs”, and expanded on their thinking by saying “I thought, oh healthy eating is a massive part of sports [please note that association of sports again]. Like the food circle, like fruit and carbohydrates”. Another participant who also included images of food on their board explained that they wanted to show “what you need to eat while you are at the gym or anything. Things like smoothies and that”. Some young people went further and categorised foods as either good and bad or healthy and unhealthy. One young person said “try not to eat bad things”, another suggested that they had selected “a list of what healthy foods you should eat and what unhealthy foods there are”, whilst a third suggested that it was important to “eat the right things when you are younger”.

This association between eating and health and labelling foods as good and bad is not new. In their work on children’s ideas about health, Burrows, Wright and McCormack [19] concluded that young people had divided the world of food into right and wrong. The danger with such a dichotomy is that young people may become obsessive about eating only foods they perceive to be good or healthy [61,62]. Equally, they might believe that any consumption of so called ‘bad foods’ may potentially have an adverse impact, including on their body image. More worryingly, in some cases it can lead to the development of, or the exacerbation of eating disorders such as anorexia [62]. Either way, such labelling of foods in this way is clearly overly simplistic and could negatively impact young people and their health.

The interview data furthermore confirmed that the young people considered healthy eating to be an integral part of physical activity. In fact, the young people seemed to perceive nutrition and healthy eating as synonymous with being physically active. Given this, it is important for educators to remember that not all messages surrounding a healthy lifestyle are interpreted and adopted by young people as intended. Furthermore, it is important that educators refrain from dichotomies and using terms such as right or wrong when discussing foods, but instead promote an overall balanced diet where everything in moderation is acceptable. 

### 3.3. Mental Health, Friendship and Getting Out 

#### 3.3.1. Mental Health

During the interviews, some of the participants spoke of not only the physical benefits of physical activity but also spoke openly about the mental benefits. One participant commented “obviously a healthy lifestyle is best for like your mental health”. Another reasoned that “ensuring you are in a good state physically and mentally, yeah like exercise can help you mentally and not just physically”. The interviews were conducted during the first COVID-19 lockdown in the UK and, although freedoms were restricted during this period, daily physical activity was still encouraged [63] in recognition of its importance to both physical and mental health. There was thus increased attention afforded to the role of physical activity in improving mental health and wellbeing at this time. This could potentially explain why some of the young people were aware of, and highlighted, its importance in this regard. Another reason may have been because they had previously learnt about the links between physical activity and mental health in school or from family. Whilst links between sport and healthy eating were expected, and have a long legacy in PE, the link between mental health and physical activity for health in schools has been less well articulated [36], at least until recently, with the physical benefits of exercise arguably being given more emphasis in PE [64].

The mental wellbeing of young people is as crucial as their physical health, as good mental wellbeing provides individuals with the resilience to face adversities in life [65]. Yet, data suggests mental health disorders in 7–16-year-olds rose from 1 in 9 in 2017 to 1 in 6 in 2020 [66]. With growing concerns about young people’s mental health, PE (and the various physical activities and sports in which young people engage in and through PE), has been viewed as one of the key ways through which to contribute to students’ mental health [67]. This has been reinforced by the Association for Physical Education (afPE) [68] in the UK [68] (p.7) who contend: 

Health and wellbeing [in PE] should be viewed holistically to comprise physical, psychological/mental and social aspects of health.

While focusing on the physical benefits of physical activity in PE is inevitable given the physical nature of the subject, greater attention could be given to the benefits of physical activity on mental wellbeing. For example, by explicitly addressing the role of physical activity in improving mood and reducing the effects of depression and anxiety within the PE curriculum [69,70,71].

#### 3.3.2. Friendship

When asked what the images chosen meant to them (see Figure 3), the young people made reference to being with friends, or taking part in activity with their friends. One participant said “well I’d say like how I would feel, I like spending time with my friends”, and another explained that they “thought like active for me is like going out with your friends and like have fun with your friends.” When describing their images on the board, one young person explained that “they represent friendship. I mean it is the type of stuff I would do with my friends”. It was clear that for some of the participants physical activity for health involved a social element and had social benefits. Again, this may have been due to the fact that at the time, their usual social interactions with friends had been restricted due to COVID-19. Whatever the reason, it is clear that friendship had a part to play in the physical activity for health experiences of some participants. 

The association between physical activity for health and friendship is increasingly being recognised in the literature. Martins et al. [72] investigated the influences of friends on physical activity during adolescence and found friends to have a positive influence on an individual in domains such as physical activity. Friendships, as Holt-Lunstad [73] and Sawka et al. [74] have argued, have the potential to be a significant part of an individual’s social experience, which can have positive implications for health as children spend a considerable amount of their time in friendship groups outside of education. Holt-Lunstad [73] (p. 239) noted that “friends may influence health by encouraging, modelling, or promoting norms of healthy behaviours” (e.g., physical activity). The participants in our study mentioned that for them, going out with friends constituted being active. These findings are important in that PE has not traditionally espoused opportunities for social interaction, nor the promotion of mental health and wellbeing [36]. In fact, Kirk [36] argued that the promotion of mental health and wellbeing would require a reordering of PE and constitute a radical change in many schools. Nonetheless, and as Parris et al. [75] found, the rapid move to online teaching during the pandemic saw a shift in practice in PE towards more of a focus on activities typically associated with mental health and wellbeing (e.g., stretching and yoga). Regardless of traditions, Martins et al. [72] (p. 9) argued that in “having a good insight into students’ friendships, relationships, and social preferences, practitioners might be in a better position to capitalize on the power of these friendships” on physical activity participation. Having such knowledge would enable PE teachers to take this into account when planning the curriculum and organising physical activity and learning opportunities during lessons, thereby helping to encourage and enhance students’ physical activity participation. 

#### 3.3.3. Getting Out

In their work on social media in developing young people’s health literacy, Dudley et al. [76] suggested that young people make extensive use of such platforms and the associated devices to interact with friends and access health information. During the interviews, some young people made various references to the need to step away from their devices and go outside. One young person said:

Instead of being on a screen you can be more out there and enjoy your life, loads of people are obviously on the Internet now, so people don’t really go outside, but you feel so much different when you go outside. You go outside it’s better and you enjoy more. Similarly, another participant reflected: 

I feel like sometimes we are so on our phones we forget to oh look outside, like we are missing that like outside, we need to help more, we should go outside. 

These young people recognised the value inherent in taking a break from technology and their devices to spend time doing other things. 

Furthermore, as can be seen from some of the images the young people selected and pinned on their boards (see Figure 4), they depicted popular activities like running and yoga being performed outside rather than indoors. Similar findings were recorded in a study of young people (aged 12–19 years) in Canada which explored their understandings of health through the use of photovoice and interviews [77]. The authors concluded that the youth portrayed the outside environment as a metaphor for good health, in which they equated being outside with being healthier.

Whilst the similarities between Woodgate and Skarlatto’s [77] research and this study are encouraging, we perhaps need to remain cautious about these findings given the timing of the study. The interviews were conducted when lockdown restrictions in England were beginning to ease and leisure centres and parks were beginning to re-open. It is thus possible that the young people’s responses may have been influenced by messages about the value of being outside and the freedom and novelty of being allowed to use outdoor facilities and spaces again following a prolonged period of lockdown. 

## 4. Limitations

There were a few limitations to this study. Firstly, as the digitally-based activity was completed on a social media platform which is powered by algorithms, the content could have been manipulated by the site’s algorithmic formula. Precautions were taken to mitigate against the influence of algorithms, including setting up individual accounts for each participant and clearing each tablet after use, but the same internet address was used at each school. In future, it is recommended that for research of this nature, individual internet protocol (IP) addresses be used for each participant rather than using one wireless address or internet dongle. In this way, incidences of the algorithms recognising similar enquiries and populating the same information to each participant would be reduced. 

Secondly, the number of participants in the study (n = 14) was lower than had been planned and was restricted with the limited access to schools and young people at the time due to the pandemic. It should therefore be acknowledged that the understandings presented here are based on those of a limited number of participants and other young people may have had different understandings.

## 5. Conclusions

The purpose of this paper was to explore young people’s understandings of what it means to be physically active. The research found the young people’s understandings in this study to be narrow and limited. Specifically, some participants had a restricted interpretation of physical activity, often referring to it as sport, and also seemed to conflate physical activity and health with body shape and weight. These findings concur with those of previous studies which have also reported young people to equate physical activity with sport and to judge health by body shape and size. They could also be partly attributed to limitations in the students’ learning and experiences with respect to physical and health in the PE curriculum. A noteworthy finding was that the young people in this study highlighted the positive effect of physical activity on mental health and also recognised the social benefits that come from participating in physical activity. This is in contrast to other studies which have found young people to overlook or fail to recognise the mental and wider benefits of physical activity participation. This suggests an increased awareness and appreciation by the young people of the benefits of activity participation on their mental wellbeing. Worthy of note also was the recognition by some of the young people of the need to step away from technology and their devices and embrace the outdoors, with physical activity affording a good opportunity for this. Although these latter findings may reflect the fact that the study was conducted during the pandemic, there is nonetheless no reason why this raised awareness of the role and wider benefits of physical activity in promoting positive mental, social, as well as physical health, should not be sustained. 

If the findings from this study are representative of other young people’s understandings, which previous research suggests they are, at least in part, they have various implications for the promotion of physical activity and healthy lifestyles within PE. For example, PE teachers need to challenge narrow and limited conceptions and conflations regarding what it means to be physically active and healthy and to educate young people that physical activity is broader than sport and encompasses a wide range of activities. PE teachers should also promote and capitalise on the wider benefits physical activity affords including the mental and social as well as physical, and draw on young people’s friendships and friendship groups as a way to boost their participation. Given the increased recognition of these wider benefits by the young people in this study, it would seem particularly timely to do this. 

More broadly, the COVID-19 pandemic clearly had an impact on young people and their freedom to be physically active. While many of these impacts have tended to be viewed negatively (i.e., limited time to be physically active and the move from face-to-face, outdoor activity, to technology-based interactions and indoor physical activity as PE), there may have been some positive gains. It seems the young people had the chance to reflect on what is important to them in terms of their physical activity participation. They acknowledged the importance of friends and the outdoors and the need to ‘step away’ from their devices and take time for their mental health. These are outcomes to celebrate and nurture, and the responsibility for sustaining these gains lies in the hands of PE teachers. It is important that the field of PE works to ensure that these gains are not lost as we support young people to be physically active in a post-COVID-19 world. 

## Figures and Tables

**Figure 1 ijerph-20-05880-f001:**
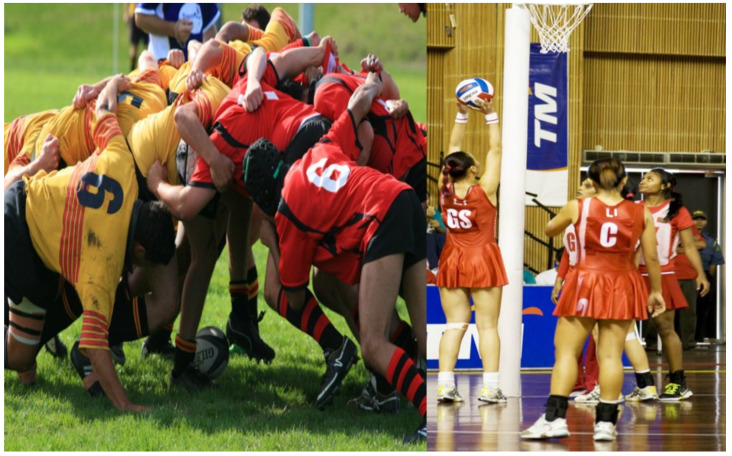
Example of sports images selected by participants (source: https://www.dreamstime.com/, accessed on 18 May 2023).

**Figure 2 ijerph-20-05880-f002:**
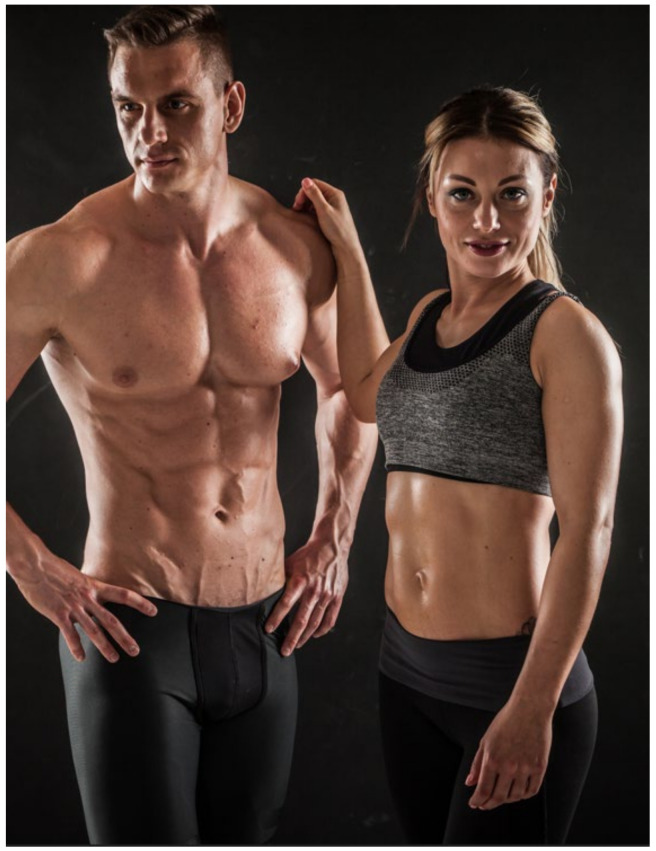
Example image from the young people’s Pinterest boards (source: https://www.dreamstime.com/, accessed on 18 May 2023).

**Figure 3 ijerph-20-05880-f003:**
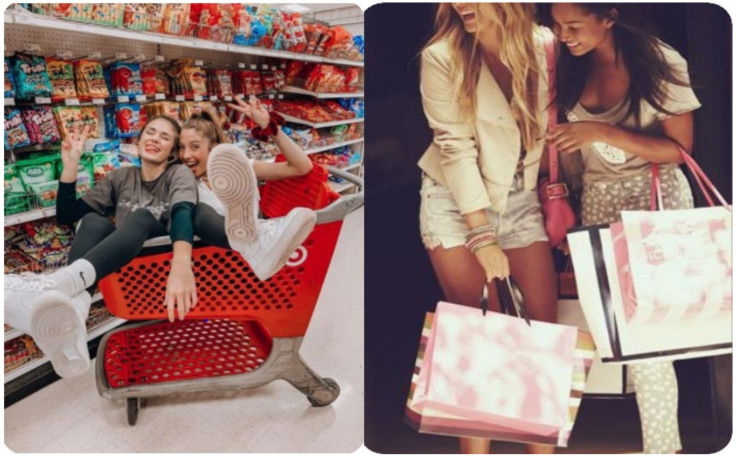
Examples of friendship images selected by participants (source: https://www.pinterest.co.uk/, accessed on 2 February 2022).

**Figure 4 ijerph-20-05880-f004:**
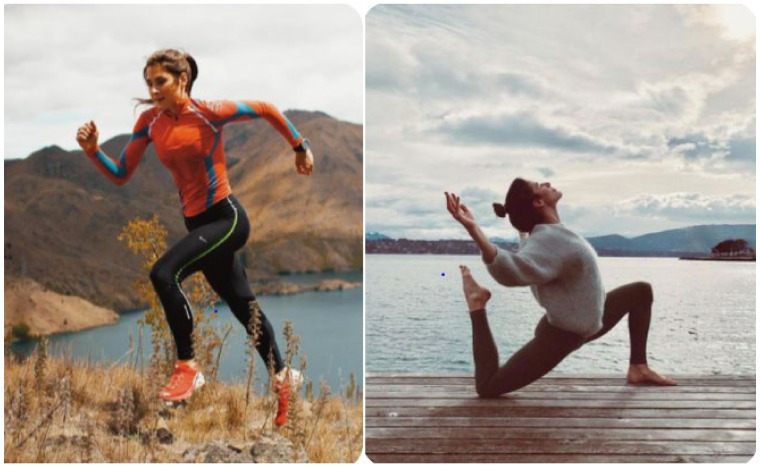
Examples of images selected that represented getting out (Source: https://www.pinterest.co.uk/, Accessed on: 2 February 2022).

## Data Availability

The data presented in this study are available on request from the corresponding author. The data are not publicly available because the larger project from which the data are drawn is still in progress.

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
