# Peer review of "“Instead of Being on a Screen You Can Be More Out There and Enjoy Your Life”: Young People’s Understandings of Physical Activity for Health"

_ijerph, 2023, doi:10.3390/ijerph20105880_

Round 1

Reviewer 1 Report

This study is very interesting, and I agree with the author's research. However, minor revisions are required.

1.     Abstract: The specified number has been exceeded, please modify.

2.     Introduction: Why is the introduction divided into two paragraphs? “Instead of being on a screen you can be more out there and enjoy your life: Young 31 people’s understandings of physical activity for health” and “Digital Technologies, Social Media and Physical Activity for Health Knowledge”.

3.     Introduction: From this introduction, the purpose of this study is not very clear.

4.     Materials and Methods: How many schools were included in this research sample?

5.     Results and Discussion: Suggest adding data or graphs to explain.

6.     Conclusion: overall well written.

7.     Please add image 1.2.3.4 Source.

Reviewer 2 Report

Dear authors,

The present manuscript has certain limitation, like:

- low number of participants can not lead to overall scientific conclusion

- presented study could have been conducted entirely online so pandemic period would not affect it

- the conclusion have no scientific truth and there are no numbers to support those conclusions, no graphs or scales

- presenting certain images to subjects can lead to limiting creative thinking so the answer can sometimes be suggested through those images

- what results were obtained after the interview with each subject and can those results be graphically represented?

- how many schools were included in the present research giving the limited number of participants ??

Best regards,

Reviewer 3 Report

Thank you for submitting this interesting article. Let me positively highlight the idea to study young people’s understandings of physical activity for health. In general, my opinion is that this study is well written and is meaningful, but some “technical” issues of the manuscript should be reconsidered and resolved.

 Abstract:  

In your abstract are 285 words, and should be up to 200 words maximum, as well as, with the required division of abstract on 4 numbered subsections (background, methods, results, and conclusion). Please, look at the IJERPH template: https://www.mdpi.com/files/word-templates/ijerph-template.dot. You have placed few keywords that are repeated from the article title (Physical Activity, and Health). There could be from 3 - 10 of them (according to the instructions), and my recommendation is that they should not be same as the words used the title. As well, you could add some keywords to the present ones.

 Introduction

Introduction it was well written and the background of this study was quite well described and explained. Title Introduction of the manuscript section should be numbered as is presented into the IJERPH template (1. Introduction). It could be divided into two subsections (1.1. Subsection, and 1.2. Subsection) but it should be done accordingly to the IJERPH template. All references should be numbered in order of appearance and indicated by a numeral or numerals in square brackets—e.g., [1] or [2,3], or [4–6].

 Materials and Methods

In general, materials and methods were well chosen and described, and in the detailed manner. Title Materials and Methods of the manuscript section should be numbered as is presented into the IJERPH template (2. Materials and Methods). It could be divided into four subsections but they should be divided by subheadings (2.1. Subsection, 2.1.1. Subsubsection, 2.1.2. Subsubsection, and 2.2. Subsection) accordingly to the IJERPH template.

 Results and Discussion

The results and discussion section was made good and in a meaningful manner, but should be just “technically” changed for the subsections accordingly to the IJERPH template. In that template, results and discussion section was positioned separately, but I understand and accept the way you have presented and explained the results. 

 Conclusions

The conclusions section was made very well and the conclusions were well drawn from the study results.

 Author Contributions: Contributions of the authors could be reconsidered accordingly to the IJERPH template and to the linked CRediT taxonomy (https://img.mdpi.org/data/contributor-role-instruction.pdf).

 Institutional Review Board Statement: That part of the additional information is missing in the manuscript (accordingly to the IJERPH template). You might choose to exclude this statement if the study did not require ethical approval.

 References

By my opinion, many references should be technically changed accordingly to the Back Matter instructions for authors (https://www.mdpi.com/journal/ijerph/instructions) and to the IJERPH template (although some differences could be found between these two instructive materials). In you use EndNote or ReferenceManager or Zotero, for more information look at https://www.mdpi.com/authors/references.

Reviewer 4 Report

My comments are listed below.

Abstract: In the abstract, the role of physical education in schools is discussed, while this research is basically not related to physical education in schools in terms of research method. Therefore, the abstract should be written based on the current research and generally discuss physical activity.

Introduction: Similar to the abstract, in the introduction there are also discussions about physical education in school and the role of social networks and digital technologies. In my opinion, the introduction of research should focus on physical activity and children's attitude towards its benefits and how they participate in physical activity.

Another important point is that in the introduction, previous studies on children's attitude towards participation in physical activity and its benefits is less mentioned. In this regard, studies that used interviews should be mentioned.

Method: As mentioned, the number of subjects is very small. However, the inclusion and exclusion criteria should be reported.

Results: The results of the study are really interesting. It is suggested to put the results of the interviews in a table so that the results can be seen together at a glance.

Discussion and conclusion: In the discussion, previous studies are less mentioned. What are the similarities and differences between the results of your study and previous studies? Can scientific arguments be made for them?

With respect,

Round 2

Reviewer 2 Report

Dear authors,

Thank you for providing a response to my questions, but I still think there are some issues that must be clarified and included in the manuscript. First, if there were 2 schools involved in the study, 14 subjects is a very low number. You say "the number of participants doesn’t detract from the richness of the young people’s voices nor the value of their voices" - but there are many more that have different opinion about physical activity...

Second, again, COVID or non-COVID, this study could have been conducted entirely online without removing the participants from the pre-COVID period. I belive that 14 subjects do not represent young people's voice and conclude that young people’s understandings about physical activity is narrow and limited...

Thank you,

Reviewer 4 Report

Dear authors,

Thank you for you revision.

I am satisfied with this version of the paper.

Best,

Author Response

Thank you for your comments